# Long-Term Fairness Inquiries and Pursuits in Machine Learning: A Survey of Notions, Methods, and Challenges

**Usman Gohar**                                         *ugohar@iastate.edu*
*Iowa State University*

**Zeyu Tang**                                           *zeyutang@cmu.edu*
*Carnegie Mellon University*

**Jialu Wang**                                          *faldict@ucsc.edu*
*University of California, Santa Cruz*

**Kun Zhang**                                           *kunz1@cmu.edu*
*Carnegie Mellon University & MBZUAI*

**Peter Spirtes**                                       *ps7z@andrew.cmu.edu*
*Carnegie Mellon University*

**Yang Liu**                                            *yangliu@ucsc.edu*
*University of California, Santa Cruz*

**Lu Cheng**                                            *lucheng@uic.edu*
*University of Illinois Chicago*

**Reviewed on OpenReview:** *https://openreview.net/forum?id=mYi6EWvFlR*

## Abstract

The widespread integration of Machine Learning systems in daily life, particularly in high-stakes domains, has raised concerns about the fairness implications. While prior works have investigated static fairness measures, recent studies reveal that automated decision-making has long-term implications and that off-the-shelf fairness approaches may not serve the purpose of achieving long-term fairness. Additionally, the existence of feedback loops and the interaction between models and the environment introduce additional complexities that may deviate from the initial fairness goals. In this survey, we review existing literature on long-term fairness from different perspectives and present a taxonomy for long-term fairness studies. We highlight key challenges and consider future research directions, analyzing both current issues and potential further explorations.

## 1 Introduction

As Machine Learning (ML) algorithms assume increasingly influential roles in high-stakes domains traditionally steered by human judgments, an extensive body of research has brought attention to the challenges of bias and discrimination against marginalized groups (Mehrabi et al., 2021; Cheng et al., 2021b). These issues are pervasive and manifest in different settings, including finance, legal (e.g., pretrial bail decisions), aviation, and healthcare practices, among others (Gohar et al., 2024; Barocas et al., 2023). The community has led tremendous research efforts toward measuring and mitigating algorithmic unfairness (Mehrabi et al., 2021). However, the majority of such fairness approaches quantify unfairness based on predefined statistical or causal criteria, often assuming a constant environment and system, defined as *static fairness*. This myopic conceptualization of fairness usually focuses on *short-term* outcomes and assesses only the instantaneous im-

Figure 1: A high-level taxonomy for long-term fairness, including notions, common modeling frameworks, and fair learning.

pact of interventions at a single snapshot, overlooking system dynamics (*dynamic fairness*) and/or *long-term* consequences over a time horizon (Liu et al., 2018; Zhang et al., 2020).

Recent works have brought attention to the misalignment of long-term fairness considerations with the fairness objectives optimized in static settings, and have shown that imposing static fairness criteria often does not guarantee long-term fairness and may even amplify discrimination (Liu et al., 2018; Hu et al., 2019). One way static fairness approaches fall short is by failing to account for the future implications of current decisions on individuals or groups, undermining their effectiveness. For example, in predictive policing, Ensign et al. (2018) observed how an initial higher allocation of police in a specific area leads to more reported incidents, perpetuating increased surveillance and exacerbating the marginalization of those communities over time. To this end, a large body of work has proposed various methods to measure, mitigate, and achieve different aspects of fairness in the *long-term* setting rather than enforcing fairness for a single time step.

Simply put, *long-term fairness* constitutes the settings outside of *static fairness* framework and *short-term* outcomes by addressing fairness over extended periods, rather than focusing solely on immediate outcomes. Importantly, it represents a spectrum of temporal fairness, from single-step effects (non-static) to sequential and multi-step fairness over extended periods, all of which are incorporated within our taxonomy (Figure 1) to capture the rich complexity of real-world scenarios. While *dynamic fairness* aligns with this concept by considering evolving dynamics over time (Li et al., 2023), long-term fairness has a much broader scope. This umbrella term has different facets, including *sequential fairness* (where sequential decisions impact fairness) and fairness over multiple time steps, among others (as depicted in Figure 1). In this work, we aim to unify the different strands of literature on long-term fairness under a common framework. Through this effort, we identify various dimensions and settings of long-term fairness and provide a rich review of existing research to explore diverse approaches and considerations for achieving fair outcomes in dynamic and non-static contexts.

**Difference from existing surveys.** Several extensive surveys on ML fairness have been conducted. A subset of these surveys (Chouldechova & Roth, 2018; Mehrabi et al., 2021; Finocchiaro et al., 2021; Pessach & Shmueli, 2022; Gohar & Cheng, 2023) predominantly focus on static fairness or a specific vision of fairness (e.g., intersectional techniques), with only brief and cursory discussions on long-term fairness in some. Meanwhile, other works have incorporated some aspects of long-term fairness. Gajane et al. (2022) reviewed fairness approaches in Reinforcement Learning (RL), with overlap limited to very brief coverage of RL techniques for long-term fairness. Separately, Patro et al. (2022) critically examined the literature on Fair Ranking, discussing the limitations of static settings and exploring modeling frameworks, like simulations, within Ranking Systems only. The survey by Zhang & Liu (2021) focused only on sequential decision algorithms, representing one facet of long-term fairness inquiries, without a critical comparative analysis of definitions, methods, and frameworks. Finally, Tang et al. (2023b) reviewed the fairness concepts in ML, drawing connections to moral and political philosophy. They provide overviews of modeling frameworks and choices in dynamic settings without making them the primary focus. In comparison to these works, our survey specifically concentrates on long-term fairness and offers a comprehensive and up-to-date overview of the field, introducing the taxonomy encompassing long-term fairness notions, mitigation techniques, and evaluation methods. Additionally, we critically examine existing techniques, highlighting key challenges and open research questions. Our literature search methodology is detailed in Appendix A.1.

**Contributions and Outline.** Our main contributions are: (1) proposing a taxonomy (Figure 1) for long-term fairness, including notions, problem settings, mitigation approaches, and modeling frameworks, (2) examining long-term fair learning methods in multiple standard problem settings and evaluations, and (3) highlighting key challenges and future directions. The rest of the paper is organized as follows: Section 2 introduces long-term fairness definition, problem settings, and modeling frameworks. Section 3 discusses fairness notions, while Section 4 covers mitigation techniques in dynamic settings. Section 5 outlines application-specific works, and Section 6 explores alternative long-term fairness settings. Next, Section 7 presents a critical analysis of the literature, including methodological assumptions, and practical implications for evaluating long-term fairness.. Finally, Section 8 discusses open challenges and future directions.

## 2 Long-Term Fairness in ML: An Overview

In this section, we first attempt to define long-term fairness and then proceed to detail the different but intricately woven aspects that form this concept.

### 2.1 What Is Long-Term Fairness?

Given the dynamic nature of real-world scenarios, long-term fairness emerges as a concept focused on achieving and maintaining fairness over a period of time rather than within isolated decisions or instances (Hu & Zhang, 2022). Long-term fairness has been described using various terms, often used interchangeably, including temporal, sequential, non-static, and dynamic fairness. However, these encompass different aspects of the overarching concept under complementary but orthogonal settings. Therefore, our first contribution is to identify and define these problem settings within a unified high-level taxonomy (Figure 1) for understanding the literature on long-term fairness. The taxonomy does not aim to enumerate or exhaust all possible combinations across the three dimensions identified, but rather serves as a flexible organizational scaffold for situating diverse approaches in the literature, with more specific interactions detailed in Figure 2 and Table 1.

### 2.2 Problem Setting

Long-term fairness was first studied within pipelines, where multiple decisions are required. For instance, a two-stage hiring model involves a candidate selection from the applicant pool, followed by hiring decisions (Dwork & Ilvento, 2018; Bower et al., 2017). However, long-term fairness has since expanded into a multifaceted concept, encompassing various problem settings. In Figure 1 (red box), we identify the four key problem settings that underpin inquiries into long-term fairness. **Dynamic Modeling** is the primary setting where the majority of the long-term fairness considerations have been explored. It addresses dynamic factors in the environment to maintain fairness over time, including cases of sequential fairness where fairness isn't achieved over multiple decision rounds due to evolving system dynamics or the failure of static fairness notions when used at each time step (Chouldechova & Roth, 2018; Cheng et al., 2021a; 2022; Yin et al., 2023). It explicitly represents the temporal evolution of a system, where the current state depends on both past states and actions, as well as feedback from the environment or population. Another facet that captures the real-world dynamics is the **Distribution Shift** setting, where deployment (testing) environments systematically differ from the training environment. This phenomenon can arise due to various factors, such as data non-stationarity (Barrainkua et al., 2023).

**Delayed impact** is a special setting of long-term fairness where the impact of the decisions is not immediately observable but manifests gradually or after a significant time lag (Liu et al., 2018). It focuses on a model of delayed outcomes, without which one cannot foresee the impact a fairness criterion would have if enforced as a constraint on a classification system. This framework is distinct because it emphasizes the lag between decisions and their eventual effects, typically *without* modeling ongoing or complex feedback from the environment. Finally, **Performative Prediction** is a recently introduced framework that formalizes scenarios involving the specific *causal* influence that predictions have on their intended targets (Hardt & Mendler-Dünner, 2023). Unlike dynamic modeling, which models how predictions influence future states over time, it abstracts away temporal dynamics and instead models the data distribution as a function of the deployed model itself. The goal is to find a *performative equilibrium*, a model that remains optimal even

after influencing the data it observes. Perdomo et al. (2020) describe it as a form of *distribution shift*, but it uniquely focuses on distribution shift resulting *solely* from model deployment. While the impact of decisions on the environment has been explored in the dynamic modeling setting, performative prediction operates as a distinct framework (see Section 6). Apart from distribution shift, all these settings intersect with various techniques like bandits, reinforcement learning, performative prediction, and strategic classification, and hence, are not clearly separable, making it challenging to write this survey and develop the taxonomy. To address this complexity and capture these interactions, we identify common modeling frameworks that accommodate these dynamics while offering flexibility in inquiry settings, as detailed below.

### 2.3 Dynamic Modeling For Long-Term Fairness

The study of long-term fairness dynamics in literature involves various approaches shaped by interactions with the environment and feedback mechanisms. Typically, these works (1) propose a *dynamics model* tailored to a specific domain, (2) expose unfairness from prolonged use of biased policies, and (3) suggest a "fair" policy to mitigate biases. As such, *modeling frameworks* (Figure 1, green box) play a pivotal role in addressing the long-term fairness challenge. In this section, we outline commonly utilized frameworks for modeling dynamics in long-term fairness.

- ***Pólya Urn:*** The Pólya Urn (Mahmoud, 2008) model is a uniquely powerful framework for capturing self-reinforcing dynamics. It simulates a stochastic process of drawing balls from an urn that increases the probability of observing certain events. At each time step, a ball is drawn and replaced with two of the same color, capturing the rich-get-richer dynamics (Akpinar et al., 2022). Its strength lies in modeling how small initial advantages can compound over time, while remaining mathematically simple, analytically tractable (Helfand, 2013), and highly interpretable. As a result, the Pólya urn is especially well-suited for studying cumulative advantage and preferential attachment in contexts like resource allocation, social networks, and recommendation systems (Barabási & Albert, 1999).

- **One-Step Feedback:** It explores the impact of decision-making on the underlying population after only a single step of feedback, distinct from works that capture feedback over multiple time steps. It considers the dynamic response of the population at deployment, influencing fairness without model updates. This single-step evaluation is driven by practical considerations, acknowledging that other variables may dominate over extended periods in real-world scenarios (Liu et al., 2018).

- **Reinforcement Learning:** Long-term fairness involves the model's interaction with the (possibly unknown) environment over time, a dynamic well-suited for RL (Liu et al., 2021). Previous works have applied RL techniques within both the *Bandit* and *Markov Decision Process (MDP)* frameworks to capture these dynamics. In both settings, the goal is to derive an optimal policy that maximizes cumulative rewards while meeting fairness criteria like Demographic Parity (Calders et al., 2009). This approach enables the exploration of complex scenarios, balancing the exploitation of existing knowledge (e.g., hiring from a known population) with the exploration of sub-optimal solutions to gather additional data (e.g., hiring from diverse backgrounds).

- **Causal Modeling and Reasoning:** Focusing on principled learning and reasoning about underlying data-generating processes, causality (Spirtes et al., 1993; Pearl, 2009; Guo et al., 2020) has been recognized as a unique tool for better understanding and mitigating bias in different contexts. Previous works have proposed to model and reason about causal relations among variables to characterize discrimination in static settings (Kilbertus et al., 2017; Kusner et al., 2017; Nabi & Shpitser, 2018; Chiappa & Isaac, 2018). Going beyond static towards dynamic and long-term settings, causal modeling of dynamics has also been explored in the literature to varying extents of detail (Creager et al., 2020; Zhang et al., 2020; Tang et al., 2023a).

**Other frameworks:** Apart from these commonly used frameworks, the literature has explored several less studied perspectives that capture different aspects of long-term fairness and its dynamics. A small but notable number of works examine incentive effects and strategic behavior (Hu et al., 2019), modeling how individuals may adapt or manipulate their features in anticipation of how an algorithm will respond, often

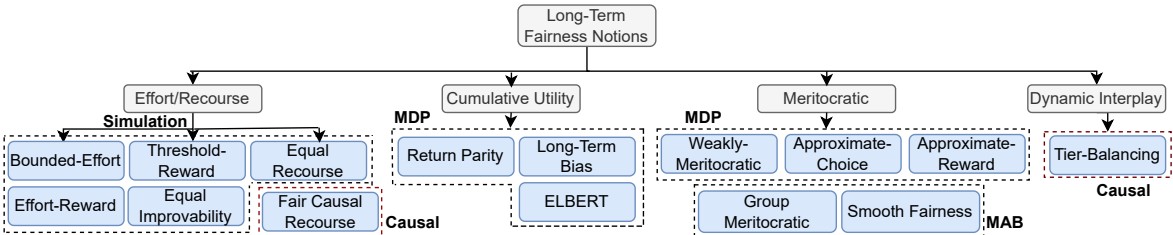

Figure 2: Taxonomy of fairness notions used in long-term settings. The dotted boxes denote the adopted modeling frameworks.

using game-theoretic or best-response frameworks (e.g., Stackelberg games) (Dong et al., 2018; Milli et al., 2019; Zhang et al., 2022). Other less common frameworks include social settings (Heidari et al., 2019), online learning (Bechavod et al., 2019), and algorithmic recourse (Ustun et al., 2019; Von Kügelgen et al., 2022). While they are not yet major themes in the long-term fairness literature, they offer useful insights for understanding the broader dynamics and directions for future work.

## 3 Long-Term Fairness Notions

Static fairness notions often focus on immediate fairness, overlooking long-term equity risks, potentially exacerbating bias over time (D'Amour et al., 2020; Liu et al., 2018). In contrast, in the long-term fairness setting, the notions capture non-static, dynamic nuances of fairness over time. In this section, we present various long-term fairness notions used in literature in different settings, as shown in Figure 2. Some of these notions can still be used in static settings, but have been adopted for long-term fairness using the dynamic modeling frameworks discussed in the previous section to capture dynamics. A summary of the main notations used in the paper is provided in the Appendix (Table 2).

### 3.1 Effort-based Fairness

Static notions of fairness predominantly center on ensuring similar outcome probabilities across all protected groups (Heidari et al., 2019; Mehrabi et al., 2021). However, these notions often operate under the assumption of a static dataset, treating individuals as immutable entities within the given data context. These approaches fail to consider that disadvantaged groups may need to exert more effort over time than advantaged groups to achieve similar outcomes, contradicting the principle of Equal Opportunity (Roemer & Trannoy, 2015; Von Kügelgen et al., 2022). This oversight persists because the distance to the decision boundary is typically greater for disadvantaged groups. Inspired by this, Heidari et al. (2019) proposed *"effort unfairness"*, which captures the disparity in average effort required by disadvantaged group members compared to the advantaged groups. Crucially, effort-based fairness is inherently long-term and dynamic since it explicitly acknowledges that individuals can adjust their qualifications or features over multiple decision cycles, and that the effort required to reach a favorable outcome may accumulate over time. This perspective allows the assessment of fairness as an ongoing process, capturing disparities that persist or compound across repeated interactions with the system.

They propose *Bounded-Effort* unfairness as a measure to quantify this disparity while constraining it within a fixed level of effort. Given a data sample denoted by a tuple $(x, y)$ where $x \in \mathcal{X} \subseteq \mathbb{R}^d$ and $y \in \mathcal{Y} = \{0, 1\}$ denote the input vector and ground-truth label, respectively. Formally[1]:

**Definition 1** *Bounded-Effort (Heidari et al., 2019). Given a constant $\delta > 0$, a classifier $f(x)$ satisfies effort fairness with $\delta-$effort if*

$$\mathbb{P}\left(\max_{\mu(\Delta x) \leq \delta} f(x + \Delta x) \geq 0.5, \, f(x) < 0.5 \mid \mathrm{g} = g\right) = \mathbb{P}\left(\max_{\mu(\Delta x) \leq \delta} f(x + \Delta x) \geq 0.5, \, f(x) < 0.5\right) \quad (1)$$

---

[1]For ease of reading and comparison, we use notations defined in Guldogan et al. (2022) for all effort-based definitions.

*holds $\forall g \in \mathcal{G}$, $\Delta x$ represents the effort (change) required to improve the feature $x$ to $x'$, $\mu : \mathbb{R}^d \to [0, \infty]$ a norm which assigns a non-negative value to input vector, and $\mu(\Delta x) \le \delta$ constraints effort to be small.*

Recognize that the condition $f(x) < 0.5$ denotes an unfavorable outcome, signifying the individual's unqualified status. In contrast, condition $f(x + \Delta x) \ge 0.5$ denotes a positive outcome, signifying successful effort ($\Delta x$) to qualify the individual. For simplicity, the features are assumed to be numerical and monotone (Duivesteijn & Feelders, 2008). The parameter $\delta$ constraints the maximum effort to ensure that individuals or groups are not required to make disproportionately high efforts for improvement. In contrast to the Bounded-Effort approach, *Threshold-Reward* unfairness (Heidari et al., 2019) shifts the focus from limiting effort to setting a lower bound on the reward. This perspective assesses how the minimum effort required to surpass a defined reward threshold varies among protected groups. Intuitively, this notion captures the idea that a disadvantaged group has to exert greater effort to get the same reward. Finally, *Effort-Reward* (Heidari et al., 2019) unfairness is defined as the difference in maximum utility (difference between reward and effort) for each protected group after additional effort, avoiding reliance on specific thresholds to prevent fairness gerrymandering (Kearns et al., 2018) and reduce result variability.

A closely related notion termed *Equal Recourse* quantifies the effort to attain desirable outcomes by measuring the average distance of individuals in different groups from the classifier's decision boundary (Gupta et al., 2019). Unlike bounded-effort unfairness, Equal Recourse aims to equalize average efforts across protected groups without imposing limits on effort.

**Definition 2** *Equal-Recourse (Gupta et al., 2019). Given the distance (effort) $\Delta x$, $f(x)$ is said to achieve equal recourse if $\forall g \in \mathcal{G}$*

$$\mathbb{E}\left[\min_{f(x+\Delta x)\ge 0.5} \mu(\Delta x) \mid f(x) < 0.5, \mathbf{g} = g\right] = \mathbb{E}\left[\min_{f(x+\Delta x)\ge 0.5} \mu(\Delta x) \mid f(x) < 0.5\right]. \tag{2}$$

One drawback of this definition is its vulnerability to outliers. Given that Equal Recourse computes the minimum required efforts by taking an *average* across all samples, the presence of outliers can distort the overall fairness measure. To address this, Guldogan et al. (2022) introduced *Equal Improvability*, akin to bounded-effort fairness, aiming to equalize average rewards for each group with a constrained effort. Formally:

**Definition 3** *Equal Improvability (Guldogan et al., 2022). Given a norm $\mu$ and a constant $\delta$, $f(x)$ achieves equal-improvability fairness with $\delta$-bounded-effort if $\forall g \in \mathcal{G}$*

$$\mathbb{P}\left(\max_{\mu(\Delta x)\le\delta} f(x + \Delta x) \ge 0.5 \mid f(x) < 0.5, \mathbf{g} = g\right) = \mathbb{P}\left(\max_{\mu(\Delta x)\le\delta} f(x + \Delta x) \ge 0.5 \mid f(x) < 0.5\right). \tag{3}$$

However, equal improvability differs from bounded-effort fairness in the conditional part $f(x) < 0.5$, i.e., it *only* focuses on equalizing the probability of *rejected* samples being qualified after applying a limited amount of effort. Von Kügelgen et al. (2022) build on the effort-based notions by introducing a causal version of Equal Recourse called *Individual Fair Causal* Recourse, focusing solely on *rejected* samples. They argue that the efforts to attain favorable decisions capture causal relations among features (Karimi et al., 2020; 2021).

**Definition 4** *Fair Causal Recourse (Von Kügelgen et al., 2022). $f(x)$ is fair if the following holds for all latent variable $u$ and groups $g, g'$*

$$\min_{f(x')\ge 0.5} \mu(x' - x(g, u)) = \min_{f(x')\ge 0.5} \mu(x' - x(g', u)). \tag{4}$$

Finally, Huan et al. (2020) extends this framework to introduce a causal-based notion that enforces equality of effort. However, it does not capture the dynamics of the environment and the model.

## 3.2 Cumulative Group Fairness

Another line of work (Chi et al., 2021; Wen et al., 2021; Yin et al., 2023) addressed the long-term nature of fairness by formalizing surrogate fairness notions grounded in static fairness notions, mirroring approaches

found in other fairness aspects such as intersectionality (Gohar & Cheng, 2023). These studies define long-term bias as the difference of the cumulative sum of group fairness (e.g., allocation rates) at each time step between different protected groups. These works have operationalized most of these notions within the MDP framework, leveraging RL techniques — a prevalent strategy for addressing long-term fairness dynamics (Zhang & Liu, 2021). An MDP is defined by a tuple $(S, A, \rho, \mathbf{T}, r, \gamma)$, where $\mathcal{S}$ is the state space (e.g., income levels), $\mathcal{A}$ is the action space (accept/reject), $\rho$ is the initial state distribution, $\mathbf{T} : S \times A \to \Delta(S)$ is the state transition function, $R : S \times A \to \mathbb{R}$ is the immediate reward function (e.g., bank's earned profits), and $\gamma$ is the discount factor. The goal is to find a policy $\pi : S \to \Delta(A)$ that maximizes cumulative reward $\eta(\pi) := \mathbb{E}_\pi \left[ \sum_{t=0}^\infty \gamma^t R(s_t, a_t) \right]$, where $s_0 \sim \rho$, $a_t \sim \pi(\cdot|s_t)$, $s_{t+1} \sim T(\cdot|s_t, a_t)$. Chi et al. (2021) propose *Return Parity*, which stipulates that MDPs from distinct demographic groups, while sharing common state and action spaces, should aim to attain roughly equivalent expected rewards under a policy. Formally:

**Definition 5** *Return Parity (Chi et al., 2021). Given distinct initial distributions $\rho_0$ and $\rho_1$ of two protected groups, $g \in \{0, 1\}$ denoting membership in the two groups with policies $\pi_0, \pi_1$ then two MDPs satisfy $\epsilon$-return parity if*

$$\eta(\pi_0) - \eta(\pi_1) \leq \epsilon. \tag{5}$$

In other words, $\epsilon$-return parity ensures that the difference in expected long-term rewards for two demographics modeled using MDPs does not exceed a specified threshold of $\epsilon$, indicating similar outcomes. The reward function can be adapted to model various fairness notions. Similarly, Wen et al. (2021) extends other fairness notions, such as Demographic Parity and Equal Opportunity, to this MDP setting of equalizing long-term rewards for different protected groups. Finally, Yin et al. (2023) follows the same principle, but instead of equalizing long-term reward, their approach focuses on minimizing the violation of the sum of static bias at each time step.

While these notions are promising for modeling fairness beyond a single time step, Xu et al. (2023b) highlights that they fail to capture the variation in temporal importance. To address this, the authors propose the Equal Long-term Benefit Rate (ELBERT) fairness notion, which considers long-term fairness as the ratio of group supply and group demand (Xu et al., 2023b). They adapt static fairness notion ratios, e.g., Equal Opportunity, by formulating them as the ratio of group supply (qualified and accepted individuals) to group demand (qualified individuals). Unlike previous adaptations of static notions, they first take the sum of group supply and group demand separately before taking a ratio. They highlight multiple cases (lending, medical treatment allocation) where temporal variations in group supply and demand lead to a false sense of fairness.

**Definition 6** *Long-Term Benefit Rate (Xu et al., 2023b). Given distinct initial distributions $\rho_0$ and $\rho_1$ of two protected groups, an MDP is said to have achieved a Long-Term Benefit Rate if*

$$\frac{\eta^{S_g}(\pi)}{\eta^{D_g}(\pi)} \tag{6}$$

*is equalized across the protected groups.*

Here, $\eta^{S_g}(\pi)$ and $\eta^{D_g}(\pi)$ represent the cumulative group supply and group demand, respectively, which are analogous to the reward function ($R$) and serve as supply $S_g(s_t, a_t)$ and demand functions $D_g(s_t, a_t)$ to the MDP at each time step.

### 3.3 Meritocratic Fairness

A body of research leverages bandit settings to model long-term fairness. Joseph et al. (2016) initially identified how no-regret algorithms (those avoiding negative feedback) in bandit settings may exhibit increasing unfairness over time. In response, they introduced an individual notion of fairness called *Meritocratic Fairness*, which posits that it is unjust to preferentially select one individual (e.g., for a loan) over another if the former is less qualified than the latter (Joseph et al., 2016), modeled in the contextual bandit setting.

In a typical bandit framework, an agent chooses an arm $A_t$ from a set of $\mathcal{K}$ arms, denoted by $k \in [\mathcal{K}]$ at each time step $t$, and receives a stochastic reward $R_t$. These rewards are assumed to be drawn from unknown

distributions specific to each arm. In a contextual setting, however, before making a decision at each time step $t$, the learner observes contextual information $x_t^j$ associated with each arm $j$. In terms of fairness, each arm is associated with a protected group, where $x_t^j$ provides information about an individual from a protected group. Each population is associated with its own underlying function $f_j$, mapping contexts to expected payoffs. Formally:

**Definition 7** *Meritocratic Fairness (Joseph et al., 2016). For any pair of arms $j, j_0$ at time $t$, if,*

$$f_j(x_j^t) \geq f_{j_0}(x_{j_0}^t), \tag{7}$$

*the algorithm is considered discriminatory, i.e., if it preferentially chooses the lower quality arm $j_0$. Conversely, the algorithm is fair if it ensures, with high probability over all rounds and for all pairs of arms $j, j_0$, that whenever $f_j(x_j^t) \geq f_{j_0}(x_{j_0}^t)$, the algorithm chooses arm $j$ with a probability at least as high as it chooses arm $j_0$.*

Liu et al. (2017) complement the aforementioned work by Joseph et al. (2016; 2018) by incorporating a smoothness constraint and protecting higher qualifications over lower qualifications in an on-average manner. Meritocratic Fairness assumes that $k$ distinct groups, each with a unique mapping from features to expected rewards, are already known. Moreover, it assumes one group member is available and always selected in each round, which is unrealistic (Joseph et al., 2018). To relax this assumption, Joseph et al. (2018) introduce the notion of *weakly-meritocratic fairness*, which does not assume knowledge of protected group membership.

**Definition 8** *$\delta$-Weakly Meritocratic Fairness (Joseph et al., 2018) A model is $\delta$-fair if, for all sequences of contexts $x_1, \ldots, x_t$, with probability at least $1 - \delta$ over the realization of the history $h$, for all rounds $t \in [T]$ and all pairs of arms $j, j_0 \in [k]$,*

$$\pi_{j|h}^t > \pi_{j_0|h}^t \quad \text{only if} \quad f_j(x_j^t) > f_{j_0}(x_{j_0}^t) \tag{8}$$

These notions, however, define fairness with respect to one-step rewards. To capture the desired interaction and evolution of the environment, Jabbari et al. (2017) extends Meritocratic Fairness to the MDP formulation, where it requires that an algorithm never probabilistically favors an action with lower long-term reward over an action with higher long-term rewards. Due to the exponential complexity of satisfying this, they propose two relaxations of this notion: 1) *Approximate-choice* and 2) *Approximate-action* fairness. Approximate-choice fairness prevents equally good actions from being chosen at very different rates, while approximate-action fairness prevents substantially worse actions from being chosen over better ones. Grazzi et al. (2022) build upon these works to propose the *Group Meritocratic Fairness* notion in the contextual bandit setting. Specifically, they introduce the idea of *highest relative rank*, which evaluates the reward's quality relative to other candidates within the same group. This approach ensures that the chosen candidates are not only selected based on their absolute rewards but also considering their position within their group's reward distribution. The aim is to address fairness disparities in scenarios with diverse group reward distributions.

### 3.4 Causality-Motivated Long-Term Fairness Inquiry

Causal modeling of underlying dynamics in long-term settings has been recognized as beneficial for analyzing social implications of automated decision-making (Creager et al., 2020; Zhang et al., 2020). To achieve long-term fairness, understanding the interplay of various processes is crucial (Tang et al., 2023b). Tang et al. (2023a) present a detailed causal model, in terms of a directly acyclic graph (DAG) for two key dynamics in decision-distribution interplay: (1) the decision-making process based on data at each time step and (2) the underlying data-generating process influencing future distributions. Rather than enforcing fairness criteria at each time step, they propose setting a goal of *achieving* long-term fairness and evaluating it alongside.

**Definition 9** *K-Step Tier Balancing (Tang et al., 2023a) A sequence of $K$ decision-making strategies $D_{T:T+K-1} := \{D_T, \ldots, D_{T+K-1}\}$ (empty set if $K = 0$) satisfies $K$-Step Tier Balancing if at the time step $T + K$ the following independence relation holds true (where "$\perp\!\!\!\perp$" denotes statistical independence, and $A$ denotes the protected feature):*

$$H_{T+K} \perp\!\!\!\perp A_{T+K}, \tag{9}$$

*where $H_{T+K}$ is induced by $K$-step dynamics.*

This notion extends beyond observed variables to include latent causal factors, i.e., "tiers", that carry on the influence of current decisions to future data distribution through the interplay between dynamics. Through a detailed causal modeling of the dynamics and the corresponding analysis of their characteristics, Tang et al. (2023a) formulate the one-step tier imbalance reduction (STIR) term and present possibility and impossibility results to illustrate the feasibility of setting *Tier Balancing* as a long-term fairness goal.

### 3.5 Discussion

Existing long-term fairness notions differ based on the modeling framework used and what aspect (e.g., effort, merit) of fairness they capture dynamically. Cumulative fairness, for instance, extends previous static fairness notions to the long-term context by aggregating fairness measures at each time step using RL. However, recurrent RL policy deployments that enforce static fairness notions at each time step may not be universally suitable for real-world scenarios (Henderson et al., 2018), necessitating the development of lightweight techniques. Recourse and effort-based notions, adapted from static settings, are only one-step analyses of long-term fairness applied iteratively to address equality of effort. One limitation is that they measure the *anticipated* effort in response to algorithmic decisions, neglecting the legal (e.g., affirmative action) and philosophical emphasis on *actual historical* effort. To bridge this gap, future work could focus on ensuring parity in outcomes based on the effort already put in and developing algorithms and datasets to capture it in long-term settings. Notably, while effort-based notions consider how individuals respond to incentives, they generally assume straightforward improvements and do not capture complex strategic adaptation, where behavior may change in unexpected or adversarial ways to influence outcomes (Zhang et al., 2022). Incorporating this strategic dimension is an important direction for future research. Moreover, some notions, like meritocratic and effort fairness, are only applicable in specific applications, such as hiring, limiting their use. While Tier Balancing is a step towards a generalized natural notion from the dynamic and long-term perspective through causal dynamic modeling, it's imperative that such notions find support in effective and efficient learning algorithms. Additionally, current notions often fail to support multiple protected groups, which fails to fully capture the nuances of fairness (Gohar & Cheng, 2023). Incorporating multiple dimensions of intersecting identities is an important challenge to solve. One interesting approach is to extend the Max-Min fairness notion (Ghosh et al., 2021) to long-term settings, optimizing outcomes for the worst-off intersectional subgroups. Finally, these notions are strictly limited to tabular data, and developing long-term fairness notions in other domains (e.g., image, video, and text) is a promising direction.

## 4 Improving Long-Term Fairness in Dynamic Setting

Several approaches have emerged to address bias in dynamic settings, leveraging interactions with the environment to maintain fairness during learning. Table 1 outlines the representative mitigation methods. Broadly, these approaches are categorized based on whether the decision-making influences the environment and, consequently, future actions. Most mitigation methods consider the dynamics in a multi-step sequential learning framework. In this setting, the system's evolution over time is explicitly modeled, with each state depending on past states, actions, and feedback from the environment. This framework captures both immediate and delayed effects, rich feedback loops, and complex interactions, and is the primary setting for mitigation methods.

### 4.1 Decisions Decoupled from Environment

In this setting, decision-making does not impact the dynamics of the environment, and most of the works assume an online learning setting where the data arrives sequentially and the model learns the dynamics on the go (Bechavod et al., 2019). The seminal work of Joseph et al. (2016) proposed a no-regret algorithm, based on chained confidence intervals, for dynamic fairness in the stochastic and contextual bandits problem for the meritocratic fairness notion. They prove that the task of learning a provably fair contextual bandit algorithm can be transformed into a "Knows What It Knows" (KWIK) learning algorithm (Li et al., 2008) and propose an efficient learning algorithm. Utilizing the MAB framework, Liu et al. (2017) propose a

| Example Works | Metric of Interest | Modeling | Time Horizon | Dynamics | Application |
|---|---|---|---|---|---|
| Liu et al. (2018) | demographic parity | - | single-step | not impacted | lending practice |
| Joseph et al. (2016) | meritocratic fairness | bandit | sequential | not impacted | - |
| Heidari & Krause (2018); Bechavod et al. (2020) | individual fairness | online learning | sequential | not impacted | - |
| Hashimoto et al. (2018) | risk disparity | online learning | sequential | not impacted | - |
| Gillen et al. (2018) | individual fairness | bandit | sequential | not impacted | - |
| Patil et al. (2021) | return parity | bandit | sequential | not impacted | - |
| Jabbari et al. (2017) | meritocratic fairness | reinforcement learning | sequential | impacted | - |
| Yin et al. (2023); Wen et al. (2021) | demographic parity | reinforcement learning | sequential | impacted | - |
| Puranik et al. (2022) | return parity | reinforcement learning | sequential | impacted | - |
| Siddique et al. (2020) | Gini social welfare function | reinforcement learning | sequential | impacted | - |
| Liu et al. (2021) | proportional fairness | reinforcement learning | sequential | impacted | ranking, allocation |
| Tang et al. (2020) | delayed impact | bandit | sequential | not impacted | - |
| Weber et al. (2022) | delayed impact | reinforcement learning | single-step | not impacted | - |

Table 1: A comparative view of long-term fairness learning methods.

Thompson-Sampling-based algorithm to achieve calibrated fairness where each arm is selected with the probability equal to that of its reward being the largest while satisfying the smoothness constraint for each round (See Sec 3.3), with a reward distribution in Bernoulli. Wang et al. (2021) further extend their work to propose a more efficient Thompson-Sampling-based algorithm by enforcing merit fairness across rounds instead of for each round and an arbitrary reward function. Heidari & Krause (2018) model the sequential individual fairness in an online learning setting. They propose a *Consistently Follow the Leader* algorithm that always picks the outcome label closest to the prediction among the labels satisfying the consistency constraint. The labels generated in such a schema are proven to follow the optimal hypothesis in bounded time steps. Also targeting individual fairness in online learning, Bechavod et al. (2020) reduce the constrained learning into the standard online batch classification problem. Thus, the common Follow-The-Perturbed-Leader algorithm can be applied oracle-efficiently to achieve sublinear Lagrangian regret. Similarly, Gillen et al. (2018) propose an algorithm where the learner aims to satisfy a fairness constraint defined using an unknown distance metric in an online linear contextual bandit setting. They assume access to only weak feedback, where a regulator can flag unfairness but cannot quantify it across individuals. Finally, Patil et al. (2021) propose a meta-algorithm, which can be applied to any MAB algorithm (e.g., UCB1) to satisfy a fairness constraint that each arm is pulled at least some pre-specified fraction of the times.

Beyond the bandit setting, Hashimoto et al. (2018) present a distributionally robust optimization (DRO) procedure to mitigate the disparity amplification over time. The core idea is to constrain the group-wise worst-case loss in each iteration; thus, the data examples with high risk are up-weighted. Specifically, let $P$ denote the distribution of data examples and $\mathcal{B}(P, r)$ denote the chi-squared ball around $P$ of radius $r$, measured by a certain $f$-divergence. The DRO loss parameterized by $\theta$ around all possible perturbation with the maximal radius $r$ around $P$ is $R_{\text{DRO}}(\theta; r) = \sup_{Q \in \mathcal{B}(P,r)} \mathbb{E}_Q \ell(\theta)$. There are two solutions to the above DRO risk. The first solution assumes the $\chi^2$-divergence and solves the dual form of the DRO risk in the following form $\hat{\theta}_\lambda = \min_\theta \mathbb{E}_P[\ell(\theta) - \lambda]^2_+$., where $\lambda$ is a dual parameter. The other solution assumes that one can compute the worst case probability distribution $Q^*(\theta) \in \arg\max_{Q \in \mathcal{B}(P,r)} \mathbb{E}_Q \ell(\theta)$ for a given parameter $\theta$ through projected gradient ascent and alternate the optimal $\theta^*(Q) \in \arg\min_\theta \mathbb{E}_Q \ell(\theta)$. The final solution $\theta^*$ and $Q^*$ will be given by the saddle point of the optimization problem. Moreover, the authors demonstrate that utilizing a one-step minimizer during each iteration reduces the online learning problem to the standard learning problem. Similarly, Hu & Zhang (2022) propose a repeated risk minimization framework to sequentially mitigate long-term fairness on post-intervention data samples.

## 4.2 Decision Impacts Environment Dynamics

A series of works (Yin et al., 2023; Chi et al., 2021) model the long-term fairness issue in the context of RL. These works typically focus on how the deployment of the decision policy influences the environment and future rewards. In this context, Jabbari et al. (2017) propose a fair polynomial time algorithm using the MDP framework to enforce the notion of approximate meritocratic fairness. Here, the distribution of individuals is considered as the state $s$ of the environment, and the decision model $f$ is considered as the action $a$. The decision model is evaluated by the risk $\ell(s, a)$ and the fairness violation $G(s, a)$ on the data observation. The utility function for the RL agent is the negative of risk and fairness violation subject to the

observed state and the selected decision policy, i.e., $r(s,a) = 1-\ell(s,a)$ and $g(s,a) = 1-G(s,a)$, respectively. The authors assume an unknown state transition probability such that $s_{t+1} \sim P(s_t, a_t)$. In the RL context, the goal of long-term fairness can be viewed as maximizing the cumulative reward $\max_\pi \mathbb{E}[\sum_{t=0}^{T} r(s_t, a_t) \mid s_0]$ while preserving the fairness violation over time $\mathbb{E}[\sum_{t=0}^{T} g(s_t, a_t) \mid s_0] \geq c$.

Yin et al. (2023) show that a modified variant of the Upper-Confidence Bound algorithm can address the long-term fairness with continuous state and action spaces. In particular, the solution is built on the assumption that the $Q$-function can be parameterized as a linear form with respect to some weight vector $w$ so that $w$ can be precisely solved by a least squares problem instead. Wen et al. (2021) propose a slightly different MDP for modeling long-term fairness. Instead of using a shared reward function for the policymaker, they assumed an additional reward function for individuals. Then, the fairness is measured as the difference of the expected individual reward between groups with the deployed model policy $\pi$. To mitigate fairness, Wen et al. (2021) propose prioritizing the policies closer to holding the fairness constraints and ranking the candidates in terms of their rewards. The sampled policies are then used to further update the decision model's parameters. The algorithm is proven to be near optimal while enforcing the static fairness constraints. On the other hand, Puranik et al. (2022) propose maximizing applicants' accumulated qualification scores with policy learning. Lastly, Siddique et al. (2020) enforces the fairness constraint that maximizes the utility of the worst-off individuals by formulating it as the generalized Gini social welfare function (Weymark, 1981). Instead of MDP, they adopt multi-objective MDPs, where rewards take the form of a vector instead of a scalar, to impose the specific notion of fairness.

### 4.3 Discussion

Compared with the standard static fairness interventions, the summarized mitigation methods mainly contribute to formulating the dynamics of human-model interactions and establishing the corresponding solutions based on the group utility shifts. Long-term fairness under unknown dynamics and/or without leveraging demographic information remains largely unexplored but promising, especially considering privacy laws (Chen et al., 2019a) that limit access to such data. From the perspective of temporal learning, it is positive that long-term fairness is focused on the multi-step sequential learning setting. However, the current research typically only focuses on optimizing the model learner at each time step. Understanding the tensions between ensuring fairness at each time step and cumulatively at the final step needs more investigation. Finally, while RL methods are flexible and learn the dynamics of the environment well, they are data-intensive (Dulac-Arnold et al., 2021; Henderson et al., 2018) and often unsuitable for real-world scenarios. Hence, developing techniques that can efficiently estimate the dynamics is crucial. Notably, the recent advance in performative prediction (Perdomo et al., 2020) provides a new perspective on off-policy learning for modeling the interplay between the model policy and agents. Finally, most current approaches do not account for individuals or groups strategically adapting their behavior in response to deployed algorithms (Estornell et al., 2023). Developing mitigation strategies that are robust to such strategic behavior remains an open challenge and is key to ensuring that fairness interventions remain effective over time.

## 5 Evaluation of Long-Term Fairness

Practical applications offer a unique context for long-term fairness considerations, depending on which one would expect context-dependent interpretations of technical findings. Therefore, in this section, we present application-specific fairness inquiries motivated by empirical scenarios.

### 5.1 Application-Specific Studies

**Credit Application.** Liu et al. (2018) focus on lending practices to quantify delayed impacts of decisions (selections) made by institutes (e.g., the bank) on different groups. They represent each group through a score distribution, where higher scores indicate a greater likelihood of positive outcomes. Assuming access to a function that provides the expected score change for individuals, they introduce a one-step feedback model and construct *outcome curves*, which describe the relation between the expected difference in the mean score after the one-step feedback and the natural parameter regimes (e.g., selection rate of the policy).

They found that both Demographic Parity and Equal Opportunity can lead to all possible outcomes, i.e., long-term improvement, stagnation, and decline, in the natural parameter regimes. This study emphasizes the impossibility of foreseeing a fairness criterion's impact without carefully modeling the delayed outcome.

**Labor Market.** The seminal work of Hu & Chen (2018) considers a two-stage dynamic reputational labor market model to capture the long-term and reinforcing nature of disparate outcomes. They propose imposing demographic parity within a temporary labor market to achieve long-term equilibrium in the permanent reputational labor market based on affirmative action. Separately, Celis et al. (2021) investigate the long-term impact of the Rooney Rule (Collins, 2007), a hiring policy aimed at mitigating implicit bias by mandating decision-makers to include at least one candidate from an underrepresented group. They employ a mathematical model of implicit bias, featuring two candidate groups, one of which is underrepresented. Each candidate possesses both a true, latent utility and an observed utility, subject to bias by a multiplicative factor. Candidates are then shortlisted by a selection panel based on observed utility. Their findings demonstrate that the rule progressively enhances minority hiring while maintaining overall utility.

**Admission-Hiring Pipeline.** Kannan et al. (2019) study a two-stage model where a student sequentially (1) gets admitted to a college, and then, if admitted, (2) gets hired by an institute. Economic literature has explored combating discrimination through affirmative actions in similar two-stage models, where students invest in human capital (at some cost) during the first stage and enter the labor market in the second stage (Foster & Vohra, 1992; Coate & Loury, 1993). The authors argue that affirmative action interventions have slow effects and advocate studying "downstream effects" for short-term impacts. They consider interventions by colleges in admission rules, grading policies, and information sharing with hiring institutes, and eliminating downstream bias in hiring decisions based on demographic group identity.

**Predictive Policing.** Ensign et al. (2018) explore resource allocation in predictive policing (Mohler et al., 2015), which deploys patrol officers based on historical crime data. They identify a feedback phenomenon present in similar algorithms across contexts like recidivism prediction and lending, where prediction outcomes heavily influence algorithm feedback. Using the Pólya urn model, they explain this feedback occurrence and propose black-box solutions, such as reweighing input incidents. Elzayn et al. (2019) go one step further by proposing an algorithm which, under simplification assumptions, converges to an optimally fair resource allocation, even in the absence of known stationary ground-truth crime rates, avoiding the runaway feedback loop effect.

## 5.2 Long-Term Fairness in Recommender Systems

Long-term fairness considerations naturally align with Recommender Systems due to evolving user interactions and item popularity, impacting system performance (Liu et al., 2021). Consequently, the RL framework is favored for its ability to model interactive dynamics. The works mostly focus on 1) *user-side fairness*, addressing equitable treatment for users, and 2) *item-side fairness*, ensuring unbiased item recommendation stemming from popularity bias due to disproportionate exposure (Chen et al., 2023).

**Item-side fairness.** Long-term fairness efforts in recommendation systems primarily target item-side fairness, addressing the challenge by modeling the sequential interactions between consumers and recommender systems using an MDP framework. A few works in this direction adopt an *Actor-Critic* RL algorithm leveraging the policy-based method (actor) with the value-based approach (critic) to learn the policy and value-function simultaneously (Grondman et al., 2012). In this setup, the actor learns an optimal policy through environment exploration, while the critic evaluates actions to determine long-term rewards (Liu et al., 2021). Utilizing this, Liu et al. (2021) propose *FairRec* that dynamically maintains accuracy and user fairness in the long term by jointly representing the user preferences (at the item and group level) as the states in an MDP model. They use the Weighted-Proportional Fairness concept, allocating resources (recommendations) proportionally to diverse entities' demands using a weighting factor. This dual-reward system strategically optimizes both accuracy and fairness in the long term. Separately, Ge et al. (2021) study the long-term fairness of item exposure, focusing on group item labels. They address the challenge of items categorized into groups based on popularity, which dynamically change with user interaction over time,

leading to shifting group labels. Using a Demographic Parity-based fairness-constrained policy optimization method, the authors propose an MDP recommendation model based on the actor-critic architecture. However, they introduce an additional critic designed for the demographic parity fairness constraint. On a related note, Morik et al. (2020) explored the rich-get-richer phenomenon in dynamic Learning-to-Rank approaches (Yang & Ai, 2021). Their solution, *FairCo*, ensures meritocratic fairness by allocating exposure to items proportional to their average relevance. FairCo integrates user feedback and dynamically adjusts its ranking based on the fairness constraint, employing a P-Controller approach for modeling, a popular feedback control mechanism (Bequette, 2003). In another work, Saito et al. (2020) tackles self-selection bias (Chen et al., 2023), where popular items tend to receive more clicks regardless of actual interest. They propose an ideal loss function optimized for relevance rather than the highest click probability. Similarly, Xu et al. (2023a) proposes an online ranking model based on the Max-Min notion of fairness in the batched bandit settings. It balances the fairness-accuracy trade-off by adding a fairness constraint and utilizing the Upper Confidence Bound (UCB) algorithm. In this regard, uniform data has also been proven to be an effective approach (Akpinar et al., 2022). This method involves randomly selecting items and ranking them using a uniform distribution. Chen et al. (2023) (Section 4.8) summarizes the various methods that use this approach. Lastly, Ferraro et al. (2021) investigates gender fairness in collaborative filtering, revealing disparities in exposure between female and male artists. They introduce an iterative re-ranking method that simulates feedback loops and penalizes male artists, resulting in adjusted rankings without significant performance impact.

**User-Side Fairness.** Relatively fewer works have focused on long-term user-side fairness. Akpinar et al. (2022) employ a simulation framework and Pólya urns to study the long-term impacts of fairness interventions in social network connection recommendations. They argue that when used in dynamic ranking settings, typical statistical notions of fairness fail to maintain equity in network sizes across different protected groups over time. This leads to minority groups having smaller network sizes and lower utility.

### 5.3 Discussion

The majority of the long-term fairness evaluations have either been context-specific (e.g., hiring) or in recommendation systems. In recommendation systems, RL methods are favored for modeling dynamics. However, a notable drawback of RL methods is their vulnerability to user tampering (Kasirzadeh & Evans, 2023), where recommender systems manipulate user opinions and preferences to boost long-term user engagement through their recommendations. Furthermore, assessing long-term fairness in a multi-stakeholder (Abdollahpouri & Burke, 2019) setting characterized by competing interests remains largely unexplored. Some other promising applications that have largely been underexplored include text, where the advent of Large Language Models (LLMs) has facilitated dynamic conversations in various settings like chatbots and healthcare assistants (Gilbert et al., 2023). Studies have indicated that model outputs can be influenced through iterative feedback (Zamfirescu-Pereira et al., 2023; Logan IV et al., 2021), raising fairness concerns. This, coupled with contextual interactions (Zhang et al., 2021), underscores the need to develop evaluation methods capable of assessing long-term fairness to capture these evolving dynamics. For instance, incorporating long-term fairness in Reinforcement Learning from Human Feedback (RLHF) (Chaudhari et al., 2024), a popular LLM optimization method, or prompt engineering to debias and capture diverse group preferences is an interesting direction.

## 6 Long-Term Fairness Beyond Dynamic Modeling

Besides addressing long-term fairness in dynamic modeling contexts, various problem settings that require distinct consideration have also tackled the problem. This section provides a brief overview of these alternative settings.

### 6.1 Performative Prediction

Performative prediction is a recently formalized framework that captures how deploying a predictive model can **causally** alter the environment or population it operates on, resulting in a model-induced distribution shift (Hardt & Mendler-Dünner, 2023). A key distinction of performative prediction is that it is often

formulated as stateless: the environment's response at each step depends directly on the model's current action, without explicit dependence on the sequence of past states. This contrasts with dynamic modeling, which explicitly tracks how the state of a system evolves over time based on both past actions and feedback loops. At its core is the concept of a distribution map $D : \Theta \to \mathcal{P}(X \times Y)$, which captures the dependence of data-generating distribution on the model parameters $\theta \in \Theta$, ignoring other reasons for distribution shift (Mishler & Dalmasso, 2022). This abstraction enables modeling a wide range of mechanisms behind distribution shifts induced by model deployment. Miller et al. (2021) propose a two-stage approach where the learner first estimates a model of the distribution map to capture the unknown parameters of the performative effects and then uses the estimated model to construct the proxy of perturbed performative loss. Next, Mendler-Dünner et al. (2020) categorize the stochastic optimization in performative prediction into *greedy deploy* and *lazy deploy*. Greedy deployment involves a sequential program that continually updates model parameters based on current data observations and deploys the updated model. In contrast, lazy deployment progressively approximates repeated risk minimization, collecting multiple rounds of observations to update the model between deployments. The abstract nature of the performative prediction framework also readily lends itself to supporting settings like strategic classification (Perdomo et al., 2020). The survey by Hardt & Mendler-Dünner (2023) offers a review of approaches, which can be extended in future work to encompass long-term fairness settings.

### 6.2 Delayed Impact

Delayed impact describes scenarios where the effects of a model or intervention appear over time, not immediately, which can turn short-term gains like bank loan approvals into long-term harms for some groups (Liu et al., 2018). It can be seen as a case of performative prediction without adopting the formal framework (Hardt & Mendler-Dünner, 2023). Importantly, delayed impact differs from performative prediction and dynamic modeling in both scope and modeling assumptions. It typically relies on simpler feedback models like one-step transitions and does not require the ongoing, high-dimensional feedback loops that characterize the other frameworks, which further simplifies both theoretical and empirical analysis. A couple of works have been brought up to mitigate the delayed impact in the context of multi-armed bandits (Tang et al., 2020) and RL (Weber et al., 2022). The proposed algorithms include two common steps to mitigate the delayed impact. In the first step, the model learns from the historical data through UCB or optimization as a classification problem. In the second step, the delayed impact will be estimated through importance sampling on the realized future data. The estimated delayed impact will be used as a reward signal that can be further used to select the arms or determine the classifier. Delayed impact is best suited for evaluating the downstream consequences of a single intervention or a limited set of interventions in relatively stable environments like college admissions and the labor market, as discussed in Section 5.1

### 6.3 Distribution Shift

An alternative research avenue explores dynamic learning environments where data distributions evolve over time. For instance, the profile of loan applicants may shift due to macroeconomic trends or changes in self-selection criteria. This dynamic necessitates continuous fairness maintenance rather than a one-time requirement. A representative work proposes a dynamic learning method that proactively corrects the classifier, estimating future data properties and retraining accordingly without direct model-environment interaction (Almuzaini et al., 2022). Within this framework, two approaches emerge 1) *Adaptive*, adapting the classifier to a known target environment, and 2) *Robust*, training classifiers to ensure fairness under perturbations when no target environment information is available (Barrainkua et al., 2023). This problem setting differs from other works due to the absence of direct environmental feedback. For an in-depth review of works in this area, we refer readers to the survey on fairness under domain shift (Barrainkua et al., 2023).

## 7 Critical Analyses and Practical Considerations

In this section, we synthesize insights from existing research to critically assess methodological assumptions and limitations and inform practical considerations.

**Questioning Normative Assumptions.** Many fairness approaches implicitly assume that "positive" decisions (e.g., loan approvals, admissions) are inherently beneficial, while "negative" decisions cause harm (Binns, 2018; Chouldechova & Roth, 2018). These underlying normative assumptions, often implicit, shape what outcomes are desirable and how fairness is operationalized in automated decisions (Cooper, 2020). However, this assumption oversimplifies complex realities, e.g., a loan approval does not guarantee a positive outcome if it leads to financial hardship. Existing long-term fairness frameworks, such as effort-based fairness notions (see Section 3.1), do not account for this nuance and tend to conflate a "positive decision" by the algorithm with a positive outcome. Moreover, long-term fairness cannot be meaningfully assessed independently of how predictions are used in policy. The social consequences of algorithmic decisions depend heavily on how those decisions are used in practice, e.g., in credit lending decisions, some policies may aim to extend loans proactively to underserved groups to promote financial inclusion, while others may adopt a conservative approach that restricts credit access to minimize defaults, which might be potentially a better long-term outcome. Zezulka & Genin (2024) demonstrate this dynamic in labor markets, showing that enforcing statistical parity and equal opportunity in risk predictions can undermine efforts to lower overall long-term unemployment and close gender gaps. These findings underscore the need to evaluate fairness through a long-term lens that accounts for downstream consequences, policy interactions, and evolving social dynamics rather than simply extending static fairness metrics or assuming that positive decisions are inherently beneficial. Therefore, discussions of long-term fairness should explicitly consider how positive (and negative) decisions distribute benefits and burdens across groups and individuals over time.

**Fragmented Landscape of Long-Term Fairness Notions.** The majority of the static fairness notions, such as statistical parity and equal opportunity, are broadly applicable across diverse domains, resulting in a unified framework for fairness evaluation. This also facilitates the development of standardized benchmarks and comparative studies across models and datasets (Ganesh et al., 2024). In contrast, long-term fairness notions are distinctly fragmented, some are even mutually exclusive, shaped by their close association with specific application domains and the diverse modeling frameworks they rely upon. For example, effort-based fairness, which emphasizes individual progress and cumulative effort, is typically applied outside of the RL framework, making it suitable for domains like education and employment where sequential decision processes or environment feedback are not explicitly modeled. This fragmentation means that these fairness notions and methods are rarely directly comparable or interchangeable, as each emerges from distinct assumptions about temporal dynamics and normative goals inherent to their respective applications and modeling paradigms. This hinders investigation of joint feasibility, or trade-offs between different long-term fairness notions, unlike static fairness literature (Friedler et al., 2019; Bao et al., 2021)

**Feasibility of the Causal Modeling of Dynamics.** In terms of principled modeling of underlying dynamics, causal models provide flexible yet precise ways to incorporate understandings of data-generating processes (D'Amour et al., 2020; Creager et al., 2020; Zhang et al., 2020; Zhang & Liu, 2021; Tang et al., 2023a). Existing works tend to (reductively) focus on observed individual-level features when capturing interplays between dynamics, but the complexity of real-world scenarios often calls for a more fine-grained description of data generating processes, potentially among both observable variables (since they are easily measured and readily available) and important latent causal factors (variables that play essential roles in practical scenario but are not particularly easy to measure, e.g., socio-economic status). Tang et al. (2023a) introduce a latent causal factor that carries over current state of affairs to future time steps when defining a long-term fairness goal, and there remains room to explore the role of additional latent factors not yet captured by the current model. Recent developments in causal discovery and causal representation learning have provided theoretical guarantees of the existence and relationship among latent factors, under mild assumptions (Xie et al., 2020; Schölkopf et al., 2021; Huang et al., 2022; Dong et al., 2024; Kong et al., 2024; Zhang et al., 2024; Zheng et al., 2025). The integration of findings from causal representation learning into current modeling approaches remains limited in the current literature on long-term fairness.

**The Gap Between Theory and Practice in Long-Term Fairness.** The current body of research on long-term fairness has established rigorous theoretical foundations, particularly through causal inference and reinforcement learning frameworks that model fairness dynamics across temporal and feedback-driven contexts (Creager et al., 2020; Tang et al., 2023a; Yin et al., 2023; Wen et al., 2021). However, practical

application remains constrained by limited and simplified empirical problem settings. This is largely due to the scarcity of longitudinal datasets annotated for fairness-relevant outcomes, challenges in capturing real-world environment dynamics (Henderson et al., 2018), and difficulties in managing confounding factors inherent in observational data (Zhang et al., 2020). As a result, empirical experiments often rely on simplistic simulations, synthetic data, and retrospective analyses, which, while useful, do not yet provide a comprehensive assessment or real-world deployment (Le Quy et al., 2022). For example, the foundational benchmark by D'Amour et al. (2020) is currently the only established benchmark to simulate dynamical systems, but it is limited in scope and based on simplified scenarios. Thus, bridging theory and practice requires the development of richer longitudinal datasets and more sophisticated benchmarks capable of modeling complex, real-world dynamics. Recent advances in large reasoning models offer a promising direction toward this goal.

## 8   Open Challenges and Future Works

Drawing on the limitations of prior work, we conclude by outlining several important research gaps and future avenues that are yet to be explored.

**The tension between long-term and short-term fairness.**   Evaluating the tension and developing techniques for achieving fairness immediately and maintaining it over an extended period presents a significant challenge. Approaches strictly oriented toward long-term goals may introduce short-term drawbacks, including reduced utility and fairness. Conversely, focusing predominantly on short-term fairness risks long-term fairness outcomes. Future work should investigate the trade-off between short-term and long-term fairness. Additionally, while the fairness-utility trade-off has been thoroughly examined in static fairness literature (Mehrabi et al., 2021), assessing this long-term trade-off and establishing benchmarks to evaluate different methods remain important directions.

**Complexity of real-world dynamics.**   As we have seen in Sections 2.3 and 5, long-term fairness necessitates modeling and characterization of dynamics. However, the complexity of real-world scenarios, including limited access to demographic data, presents significant challenges for practitioners to fully understand the interplay of various dynamics and effectively address the discrimination involved. Translating insights from studies focused on specific applications to broader contexts and bridging the gap between empirical fairness evaluations (Gohar et al., 2023) and theoretical guarantees (Tang & Zhang, 2022) are important and ongoing challenges in the field.

**Intersectional identities.**   Current long-term fairness research has primarily concentrated on single-attribute group identities, such as gender or age. However, numerous studies reveal that bias is a nuanced phenomenon that spans multiple protected groups and intersecting identities (Gohar & Cheng, 2023). This complexity poses unique challenges, notably the exponential number of subgroups. Future work should focus on developing methodologies and frameworks that capture and mitigate bias at the intersection of various attributes in long-term settings.

**Explainability over time.**   Explainability in AI has gained tremendous attention due to the need for transparency, which significantly affects acceptance within the broader community (Gross, 2007; Shin, 2021). Long-term fairness notions, such as effort-based fairness, necessitate using explainable methods. It is vital to empower the impacted population to understand how decisions are made and determine strategies to enhance their qualifications in subsequent instances. Furthermore, incorporating explainability methods in dynamic fairness-aware recommendation environments is another direction (Chen et al., 2019b).

**Long-term dynamics in other AI domains.**   Studies have shown that prompt engineering and fine-tuning in LLMs directly influence the model toward generating the desired outputs (Zamfirescu-Pereira et al., 2023; Logan IV et al., 2021). In real-world scenarios, user interactions are dynamic and may even be strategic, guiding the LLMs in tailoring responses to their needs through repeated feedback. An important research direction is developing evaluation methods that capture these dynamics and assess bias and fairness in LLM (and other large models) outputs. This, coupled with the interaction context, significantly influences

LLM's output (Zhang et al., 2021). On the other hand, it is also crucial to steer the response of LLMs to consistently align with human values represented by different demographics and cultures (Solaiman et al., 2023) in multi-round dialogues.

**Alignment with policy and regulatory framework in the long run.** *Trustworthy AI* has drawn increasing attention from policy and regulatory efforts. For example, the European Commission High-Level Expert Group on AI presented the Ethical Guidelines (European Commission, 2019) that put forward a set of 7 key requirements an AI system should meet to be deemed trustworthy. The Organisation for Economic Cooperation and Development also proposed a framework to compare implementation tools for promoting *Trustworthy AI* (Organisation for Economic Co-operation and Development (OECD), 2020). Consequently, it is important to consider long-term fairness together with other aspects of *Trustworthy AI.*

## 9    Conclusion

Recent research highlights the inadequacy of static one-shot fairness approaches in capturing unfairness over extended periods. To offer researchers an understanding of the field, we present a unifying review of long-term fairness in multiple problem settings and frameworks. First, we provide readers with a high-level overview of the conceptual frameworks employed to model long-term fairness problems. Next, we propose the first taxonomy on long-term fairness, wherein we explored long-term fairness definitions, mitigation methods, and representative evaluation techniques across various applications. We also discuss briefly other problem settings that tackle this issue. Finally, we critically dissect the literature and discuss open challenges and opportunities of long-term fairness research. Ultimately, we hope our survey serves as a foundational resource for those new to the field, fostering an understanding of the complexities associated with long-term fairness and drawing attention to the potential societal consequences that may arise without effective long-term fairness measures.

## Acknowledgments

Lu Cheng is supported by the National Science Foundation (NSF) Grant #2312862, NSF CAREER #2440542, NSF-Simons SkAI Institute, National Institutes of Health (NIH) #R01AG091762, and a Cisco gift grant. We would also like to acknowledge the support from NSF Award No. 2229881, AI Institute for Societal Decision Making (AI-SDM), the NIH under Contract R01HL159805, and grants from Quris AI, Florin Court Capital, and MBZUAI-WIS Joint Program. Zeyu Tang is supported by the National Institute of Justice (NIJ) Graduate Research Fellowship, Award No. 15PNIJ-24-GG-01565-RESS. We would also like to thank Nina Mehrabi for feedback on an earlier draft.

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

## A  Appendix

### A.1  Survey Methodology

We employed several established approaches to conduct a comprehensive search for relevant papers under the umbrella of long-term fairness. Initially, we curated a set of seed papers that have discussed and evaluated long-term fairness problems (Zhang & Liu, 2021; Liu et al., 2018; Hu & Chen, 2018; D'Amour et al., 2020; Zhang & Liu, 2021; Zhang et al., 2020). Following common practice (Desolda et al., 2021; Snyder, 2019), we used these papers and domain knowledge to identify and develop a set of keywords to search for relevant work: "long-term fairness," "dynamic fairness," "(non-)static fairness," and "sequential fairness." Primarily,

we searched proceedings of relevant conferences and journals that were likely to publish work that fit the definition of long-term fairness. These included but were not limited to FAccT, NeurIPS, AAAI, ICML, AIES, ICLR, and IJCAI. We also followed citation trails to and from these widely cited papers and querying across Google Scholar and Arxiv to identify additional related work.

| Framework | Symbols | Description |
|---|---|---|
| Classification | $f$ | A decision model parameterized by a vector $\theta \in \Theta$ |
| | $x$ | $x \in \mathcal{X} \subseteq \mathbb{R}^d$ input features |
| | $y$ | $y \in \mathcal{Y} = \{0, 1\}$ ground truth label |
| | $g$ | $g \in \mathcal{G}$ sensitive groups |
| | $\Delta x$ | Effort (change) required to improve the feature $x$ to $x'$ |
| MDP | $S$ | State Space (e.g., income levels) |
| | $\mathcal{A}$ | Action space (e.g., accept/reject) |
| | $R$ | $R : S \times A \to \mathbb{R}$ reward function (e.g., bank's profits) |
| | $\rho$ | Initial state distribution |
| | $\mathbf{T}$ | $\mathbf{T} : S \times A \to \Delta(S)$ State transition function |
| | $\gamma$ | Discount factor |
| | $\pi$ | $\pi : S \to \Delta(A)$ is policy |
| | $\eta(\pi)$ | Cumulative Reward |
| MAB | $\mathcal{K}$ | Set of arms |
| | $t$ | Time step |
| | $A_t$ | $A_t \in \mathcal{K}$ action (arm) selected at time $t$ |
| | $R_t$ | Stochastic reward at time $t$. |

Table 2: Table of Notations

