# OpenReview forum: "Long-Term Fairness Inquiries and Pursuits in Machine Learning: A Survey of Notions, Methods, and Challenges"
_TMLR — Accepted by TMLR_

### Review · Reviewer_dGLx · 2025-04-23

**Summary Of Contributions:**

**Summary:**
This paper, *"Long-Term Fairness Inquiries and Pursuits in Machine Learning: A Survey of Notions, Methods, and Challenges,"* provides a comprehensive survey of the current landscape of research on long-term fairness in machine learning. The authors review a wide range of literature, organizing existing works around key fairness notions and methodological approaches. The paper highlights the significance of moving beyond short-term metrics and emphasizes the importance of evaluating fairness over extended time horizons.

**Audience:**

Yes

**Broader Impact Concerns:**

Since the work is a survey and does not introduce novel algorithms or systems with direct deployment risks, and because it already highlights key fairness challenges, I did not identify any additional broader impact concerns that would require further elaboration.

**Claims And Evidence:**

Yes

**Requested Changes:**

**Suggestions for Improvement:**

1. Refine the distinction between short-term and long-term fairness. Consider introducing intermediate or hybrid notions that may better reflect the temporal dynamics in machine learning applications.
2. Expand the discussion on online learning approaches by including theoretical results (where available) and indicating whether referenced works are primarily theoretical or empirical.
3. Enhance the clarity of the taxonomy or structure used to organize the surveyed works to aid readers in navigating the landscape more effectively.

**Strengths And Weaknesses:**

**Strengths:**
- The paper addresses an important and timely issue in the fairness literature by focusing on long-term implications, which are often overlooked in favor of short-term metrics.
- It provides a structured overview of the field, categorizing existing work and pointing out research gaps and challenges.
- The survey includes many representative works, demonstrating that the authors have performed a thorough review of the literature.

**Weaknesses:**
- The categorization of fairness into "short-term" and "long-term" could benefit from a more nuanced treatment. As it stands, the dichotomy seems overly binary. It would be helpful to explore whether a spectrum or taxonomy of temporal fairness notions could better capture the complexity of real-world scenarios.
- The section discussing online learning as a tool for long-term fairness is promising but lacks sufficient theoretical grounding. It would strengthen the paper to elaborate on the theoretical underpinnings of the approaches mentioned.
- Additionally, when referring to existing literature, it would be helpful for the authors to more clearly distinguish between empirical and theoretical contributions. This clarification can help readers understand the maturity and foundational depth of different sub-areas.

---

> ### Author Response · Authors · 2025-05-20
>
> We thank the reviewer for their time and effort in providing valuable and constructive feedback for our work.
>
> **Response to RC1:** We agree that the distinction between "short-term" and "long-term" fairness can be overly simplistic and that a more nuanced treatment better reflects the complexities of real-world scenarios. In response to this concern, we would like to clarify that our taxonomy ( Figure 1) is intentionally structured along three dimensions to capture a more refined spectrum of temporal fairness notions. For example, the modeling frameworks we discuss range from one-step feedback models, which only consider more immediate effects, to reinforcement learning approaches that account for outcomes over multiple time steps. This structure aligns with various fairness notions in the literature, such as cumulative utility and effort-based measures, which themselves can span different temporal horizons. Our primary aim is to synthesize and organize existing work rather than introduce new intermediate or hybrid notions. Nonetheless, we have revised the introduction (blue text) to explicitly mention this
>
> **Response to RC2:**  We appreciate the reviewer’s feedback. While including theoretical results could benefit some readers, it might risk making the survey a bit overly complex. We hope our survey serves as a foundational resource for those new to the field. However, to address your feedback, we will briefly highlight key theoretical insights where relevant to guide interested readers toward more detailed studies and clarify the nature of referenced works where possible in the camera-ready (it will require some time).
>
> **Response to RC3:** Please see RC1 and our responses to Reviewer kMgW02, where we have made revisions. If there are any more concrete suggestions to improve the clarity of the taxonomy, we will definitely welcome the feedback to improve our work

---

### Review · Reviewer_kMgW · 2025-05-02

**Summary Of Contributions:**

This paper surveys the literature on various fairness methodologies that can be broadly categorized as focused on "long-term fairness", rather than static approaches that focus on a single dataset or decision. It constructs a taxonomy of these approaches, presents some in light detail, and discusses where the literature leaves space for future work.

**Audience:**

Yes

**Claims And Evidence:**

No

**Requested Changes:**

The most important change needed is a clean-up of the taxonomy to address questions like:
* Why is effort-based fairness a long-term approach?
* Why are incentive effects and strategic behavior not part of the taxonomy?
* What distinguishes "delayed impact", "performative prediction", from "dynamic modeling"? And why is this difference meaningful?
* Why does the urn model warrant special consideration over other dynamic models?

Secondly, the paper presents a lot of methods but does make the reader aware of the huge drawbacks the some have. For example,
* Causal fairness approaches cannot be deployed in real systems because the real causal DAG is never known (and the outcome of the approaches depend entirely on which stylized DAG is used)
* Many of the discussed fairness approaches assume that one decision is "positive", but in almost every important context (lending, admissions, medicine, criminal justice) neither action is straightforwardly positive or negative. This is particularly concerning in a paper that is supposed to be about the long term!

**Strengths And Weaknesses:**

I think there is value in surveying this line of research, both as a useful resource for researchers and because, ultimately, the long-term effects of our decision-making systems on people is what we should be most concerned about.

The survey covers a reasonable set of papers; nothing obvious has been left out. The relative emphasis on different papers is puzzling at times, for example, multiple "effort-based fairness" methods are defined in detail (despite these not being obviously "long-term", see below for further critique) while incentive effects and strategic behavior -- which seem extremely important to any change in high-stakes decision-making -- are mentioned only in citations.

The taxonomy proposed in this paper is functional, but only just. It pulls together disparate ideas into loose categories, and seems to simultaneously operate at too-coarse and too-specific levels of detail. For example, the Polya urn model (a specific model of one type of dynamic) is given the same prominence in the taxonomy as "Dynamic Modeling", which could plausibly encompass all the other settings in the "Problem Setting & Mitigation" category as well as all the modeling frameworks including the urn model.

In places it's hard to determine whether the summaries of various methods and their justifications are weak, or if the underlying research that is being summarized is simply underwhelming. This points to a further weakness: the discuss sections don't critically engage with the methods presented (eg "when would this method be (in)appropriate to deploy?", "are the normative assumptions/assertions underlying the method well-justified", "can this method be deployed in pratice") but instead mostly point out room for straightforward future work (eg "current notions often fail to support multiple protected groups"). But future work in an area isn't productive if the work it's building on is unworkable or normatively indefensible.

Ultimately, I think this paper is a flawed execution of a good research direction.

Further issues/notes:

> “For example, in predictive policing, Ensign et al. (2018) observed how an initial higher allocation of police in a specific area leads to more reported incidents, perpetuating increased surveillance and exacerbating the marginalization of those communities over time.”

Doesn’t this summary provide the same myopic, simplistic fairness analysis that this paper is criticizing? It assumes that higher allocations of police to an area (which might grow as the number of reported incidents increases) are bad for the people there. But we can’t conclude this unless we’ve studied the long-term outcomes of the policy on (among other things) crime, safety, subjective well-being, the local economy, etc.

I don’t think “delayed impact” is meaningfully different from dynamic modeling. The “delayed impact” in the Liu et al paper was the effect on a borrowers’ credit score from repaying (or failing to repay) their loan. But this is a mechanical function of the outcome being predicted (default/repayment), not some second-order or downstream effect. In almost every setting we care about, the impact of decisions aren’t immediately apparent: the benefits from admission to college don’t accrue until the student graduates, the benefits from cancer treatment accrue slowly as treatment proceeds, etc. So I don’t see what value “delayed impact” provides as a separate category. To me, the value of the Liu et al paper is that it was one of the first to consider _any_ impact of a decision on the decision subject, rather than (as prior work did) assuming it was good for a borrower to receive a loan even if they defaulted on it.

It seems like an oversight to study long-term fairness without considering incentive effects. Is that just because there is very little research in this area?

Every fairness notion in the first section (3.1, Effort-based fairness) assumes there is a “positive outcome” (which would be better termed a “positive decision”, since it is assigned by the classifier). But the most important insight of long-term fairness is that the decision itself does not determine whether the short- and long-term outcomes of the decision are good. It’s not a good outcome to be admitted to university if you eventually drop out with a lot of debt, and it’s obviously not uniformly good to receive chemotherapy – it’s only good if you need the treatment and it works!

Furthermore, what about these effort-based notions of fairness is long-term? Defs 1, 2, and 3 all just constrain a classifier’s mapping from features to classifications, which is also what static fairness notions do.

> “Causal modeling of underlying dynamics in long-term settings has been recognized as beneficial for analyzing social implications of automated decision-making”

In reality it’s extremely difficult to measure the causal effect of a single decision or policy on humans, let alone construct a correct DAG of interconnected causal effects over time. It’s important that any discussion of causal fairness points out this huge impediment to these methods being practically applicable.

The “Long-Term Fairness Beyond Dynamic Modeling” section starts with: “Performative prediction refers to the dynamic influence of model decisions on future data distribution.” But what is “dynamic modeling” if it’s not modeling “the dynamic influence of model decisions on future data distribution”? The Performative Prediction section did not illuminate this and didn’t explain why it deserved to be called out in its own section.

Similarly, delayed impact and distribution shift seem to be exactly the sort of dynamics one might want to model if considering long-term fairness, so it’s weird that they are in the “Beyond Dynamic Modeling” section.

---

> ### Author Response · Authors · 2025-05-19
>
> Thank you for your thoughtful and constructive feedback on our paper. Your valuable feedback is very helpful and improves the quality of our work. We appreciate the time you took to review the manuscript, and we have carefully considered your suggestions. Below, we provide a detailed response, along with the corresponding changes we have made, which are highlighted in blue within the revised manuscript
>
> **RC1: Why is effort-based fairness a long-term approach?**
>
> We understand the reviewer's concern that effort-based fairness as a standalone seems like a static fairness notion. While it is true that Definitions 1, 2, and 3 formally constrain a classifier’s mapping from features to outcomes, similar in form to static fairness, the underlying motivation and interpretation of effort-based fairness are inherently dynamic and long-term in nature. Specifically, effort-based fairness addresses how individuals or groups can **improve** their outcomes through changes to their features (evolving trait), which are often the result of sustained effort **over time**. Therefore, effort-based fairness is often studied within dynamic modeling frameworks that account for the evolution of individual features, capturing fairness in a long-term sense. As such, the application of the notion is inherently longitudinal, often accomplished in the dynamic modeling framework via simulation studies. We have further highlighted this nuance in Section 3.1 to avoid any confusion for the reader.
>
> **RC2: Why are incentive effects and strategic behavior not part of the taxonomy?**
>
> Thank you for this insightful comment. We agree that incentive effects and strategic behavior are potentially important aspects of long-term fairness. However, as noted, these factors remain relatively underexplored and are not yet major themes in the long-term fairness literature, which we acknowledge in Section 2 of the manuscript. The long-term fairness literature has *primarily* focused on temporal and feedback dynamics, examining how algorithmic interventions impact outcomes over time and across groups. Incentive effects and strategic behavior, though important, have received less attention and are often treated as complementary concerns. Accordingly, our taxonomy centers on organizing methods around these dynamic modeling and evaluation perspectives. That said, we recognize that incentive and strategic considerations represent an important axis of analysis. We have revised Section 2 to highlight their relevance (Other frameworks) and encourage further exploration from multiple angles (Notions, mitigations), including future work in Sections 3.5, 4.3

---

> ### Author Response · Authors · 2025-05-19
>
> **RC3: What distinguishes "delayed impact", "performative prediction", from "dynamic modeling"? And why is this difference meaningful?**
>
> We agree that these concepts are somewhat related, and we appreciate the opportunity to clarify their definitions and the rationale for distinguishing them.
>
> ***Performative prediction*** is a recently formalized framework that captures how the deployment of predictive models can **causally** influence the environment or population to which they are applied. The central feature is that the act of prediction itself changes the distribution of future data, i.e., it is a model-induced distribution shift. Importantly, performative prediction is often formulated as stateless: the environment’s state is modeled as a direct function of the classifier's current action, without explicit temporal dependence on previous states. This distinguishes it from dynamic modeling, which explicitly tracks state evolution over time. The performative prediction framework is gaining traction as it generalizes and subsumes other settings, such as strategic classification, and is increasingly being applied to areas like algorithmic fairness and strategic behavior. As such, we believe the framework is relevant on its own (Hardt, M., & Mendler-Dünner, C. (2023))
>
> ***Delayed impact*** describes scenarios in which the effects of deploying a machine learning model or intervention are not immediately observable but instead materialize over time. This concept underscores the temporal lag between an action and its eventual consequences, which is particularly salient in fairness analyses: interventions that appear beneficial in the short term may, upon closer examination, result in unintended disparities or negative outcomes for certain groups in the long run. Unlike performative prediction, the study of delayed impact does not necessarily require modeling feedback or causal influence from the model’s predictions; rather, it focuses on how present decisions shape future outcomes, often without complex feedback mechanisms. Furthermore, analyses of delayed impact typically employ relatively simple feedback models, such as one-step transitions, rather than the intricate, ongoing, or high-dimensional feedback loops that characterize dynamic modeling or performative prediction frameworks
>
> ***Dynamic modeling*** explicitly represents the temporal evolution of a system, where the current state depends on both past states and actions, as well as feedback from the environment or population. This allows for the modeling of rich, realistic feedback loops and long-term dependencies. Dynamic modeling is the most general of these frameworks, accommodating **both** immediate and delayed effects, as well as complex interactions between model actions and system states over time. This is the primary setting that the literature has focused on.
>
> We thank the reviewer for bringing this up. We realize that the draft could benefit from an enhanced discussion that clearly distinguishes between the frameworks and may confuse readers. We have incorporated this in our revised draft. (Sections 2.2, 6.1, and 6.2)
>
> **RC4: Why does the urn model warrant special consideration over other dynamic models?**
>
> We contend that the Pólya urn model deserves particular attention for several reasons.
>
> **1.  Unique Feedback Dynamics:** It is a special modeling framework that captures the self-reinforcing dynamic, where the probability of a particular outcome increases each time it is observed.. This "rich-get-richer" effect is mathematically distinct from the update mechanisms in standard reinforcement learning, which typically adjust action values based on rewards but do not inherently produce such path-dependent compounding effects.
>
> **2.  Analytical Tractability and Interpretability:** Urn models offer a high degree of mathematical tractability, allowing for closed-form analysis in many settings. Unlike RL, they do not require learning a policy, estimating transition probabilities, or simulating environment interactions. Instead, they model reinforcement through a straightforward stochastic process driven by past outcomes, making them both easier to analyze and more interpretable.
>
> **3. Modeling Social and Network Phenomena:** The Pólya urn and its extensions are especially well-suited for modeling social influence, collective dynamics, and network growth, where reinforcement and imitation play a central role. For example, preferential attachment in network science is elegantly captured by urn models, providing clear insights into how inequalities and hubs emerge, phenomena less naturally represented by other dynamic frameworks.

---

> ### Author Response · Authors · 2025-05-20
>
> **Response to RC: Causal fairness approaches cannot be deployed in real systems ...**
>
> Developments of the research on causal discovery over the past two decades have established various identification results. Here, by "identification," we are referring to the theoretical guarantees of the correctness of the recovered causal model (which can take the form of a DAG, or other graph families like MAG), under mild assumptions. While it could be the case that "the real causal DAG is never known" if without any assumption/knowledge about the underlying data-generating process, the claim itself does not render the whole project of causal inference (and based on which, causal fairness) inapplicable in the real world.
>
> **Response to RC: Many of the discussed fairness approaches assume that one decision is "positive"...**
>
> We totally agree that it could be highly nontrivial when interpreting positive/negative implications of the decision or action, especially in complicated scenarios. When we review the previous developments in the literature, we prioritize fidelity to the original presentations in previous works, summarizing prior literature distinctly and without conflating it with our own reflections or interpretations.
>
> In light of your comment, we have included the implications of different decisions in the discussion on the complexity of real-world dynamics. (Section 7)

---

### Review · Reviewer_EjFb · 2025-05-06

**Summary Of Contributions:**

The paper surveys work on notions, methods, and frameworks for long-term fairness. The paper taxonomizes these works along multiple axes: 1) fairness notions, 2) modeling of dynamics, and 3) mitigation methods. Outside of this taxonomy they also provide other considerations for long-term fairness. They also survey work that study long-term fairness tailored to specific applications including recommender systems. They also present a number of open questions and limitations that the current literature does not yet address.

**Audience:**

Yes

**Claims And Evidence:**

Yes

**Requested Changes:**

Some changes that would strengthen the paper:

1) Discussions on whether it makes sense to study all combinations of all dimensions of the taxonomy. Are there certain combinations that do not make sense? For example, is there a certain modeling that only makes sense for particular notions and mitigation methods?

2) Having a table or figure showing where papers fall under all three taxonomy dimensions. I see Figure 2 that relates notions of fairness and modeling. It would be nice to have another figure that also includes mitigation methods and how this relates to the other two dimensions of fairness notions and modeling.

**Strengths And Weaknesses:**

Strengths:
1) I believe the survey covers the relevant topics in the area of long-term fairness that I am aware of.

2) The taxonomy created in the paper provides a nice template to organize the papers in the field.

Weaknesses:
I feel like there could be more discussion of how the papers and methods relate to or differ from each other. Please see suggested changes for concrete examples.

---

> ### Author Response · Authors · 2025-05-19
>
> Thank you for your time and effort in reviewing our work. We have carefully studied your constructive suggestions and discuss them below.
>
> **Response to RC1:** We agree and do not suggest/claim that every combination of dimensions is equally meaningful or practical. In fact, some modeling frameworks are inherently better suited to particular fairness notions or mitigation strategies. Rather, the taxonomy is structured to provide a framework that intuitively stitches the fragmented literature and facilitates a broad understanding of long-term fairness (which we believe to be along the three key dimensions). More specific interactions and dependencies between modeling assumptions, fairness notions, and mitigation strategies are mentioned in Figure 2 and Table 1, which illustrate how certain combinations tend to co-occur in the literature. Also, we already provide some discussion of how certain settings (e.g., practical considerations, use-case, computation) dictate what notion can be used to capture the long-term dynamics in Section 3.5 (Discussion).  Nonetheless, we understand how it may confuse the reader. We appreciate the reviewer’s suggestion and have made these considerations more explicit in the revised manuscript and clarifying that not all combinations are possible in Section 2.1
>
> **Response to RC2:** Thank you for your positive feedback on Figure 2. We would like to clarify that Table 2 already includes how the mitigation methods (in the column of example works, i.e., Dynamic Modeling) relate to notions (column metric of interest) and modeling. More specifically, the mitigation methods discussed are situated within the “Dynamic modeling” setting, which corresponds to the “Problem Setting” dimension of our taxonomy. Additionally, Table 1 provides an overview of the fairness notions (the second dimension) addressed in the mitigation literature, albeit at a more granular level. However, papers in the literature often do not propose mitigation strategies for **all fairness notions**; in fact, some notions are proposed solely for characterizing longer-term unfairness without addressing mitigation. To avoid misclassification, we limited our current mapping accordingly.

---

### Comment · Action_Editor_16ki · 2025-05-06
**Summary: Need of improvement**

After reading the reviews, I feel that the review paper is generally well-received for tackling an important and underexplored topic — fairness in machine learning over extended time horizons. It offers a comprehensive overview of literature, a proposed taxonomy, and insight into future directions. However, several areas for improvement are consistently pointed out across reviews: The reviewers didn't spot any specific errors that could undermine rigor but have specific suggestions for improving the clarity. Here's are short summary:

**Taxonomy**
- Clarify terminology (e.g. what is long-term fairness vs short term fairness)
- Clarify concepts (e.g., delayed impact vs. dynamic modeling).
- Include a comprehensive figure/table mapping key papers to all taxonomy dimensions.

**Improved presentation**
- The authors have specific suggestions for adding tables, such as mitigation methods (Reviewer EjFb)

**Coverage**
- long-term fairness w/ and w/o considering incentive effects. (If there is no research, could that be part of a research agenda)
- effort-based fairness
- combinations of delayed impact and distribution shift (if there is no research, could be part of a research agenda)
- strategic behavior
[see Reviewer kMgW for more detailed information]

**Critical analysis**
- Currently, the paper often summarizes existing methods without evaluating their limitations or practical deployability. My suggestion is to more critically assess the feasibility of methods (e.g., causal inference challenges in real-world deployment).
- Discuss normative assumptions in fairness methods (e.g., presumption of “positive decisions”).
- Indicate when methods are only theoretically sound but practically challenging.
- Elaborate on the empirical vs. theoretical maturity of surveyed approaches.

=> For the critical analysis: While I leave the exact path forward to the authors, I could see different choices like a new Section "Recommendation" or "Critical analysis" or something along these lines. Similarly, some of the suggested points could also be better placed in other sections.

Additional typos from my reading
- "Fig:1" = "Fig. 1"
- Sec 3.1: the "." in the first ref is incorrectly placed, and the second ref should be \citep instead of \citet
- The problem of \citep and \citet continues in Sec 3.2 and beyond
- Why is Liu et al. (2018) listed as demographic parity, and not delayed impact? Same table: I think ranking and lending are not both applications, but one is the task, the other is the application; I also think it is not correctly listed when the task is recommendation and when not (because it sometimes benefits from different modeling approaches)
- Section 6 is called "Beyond dynamic modeling", but in one way or the other, these approaches are all with some dynamic component
- Research agenda: what about the fairness-utility tradeoff (as stipulated by https://arxiv.org/abs/2207.10991)? What about identifiability of the causal approaches (the latter could be nicely back-linked to the main paper to build up a story)?


Note: I wrote some of the comments pre-rebuttal so some may have been addressed. We used the above fornour internal discussion but somehow it was not made available to the authors. I apologize!

---

### Author Response · Authors · 2025-05-20
**Thank you to the reviewers.**

We would like to abundantly thank all the reviewers for the effort and time in providing feedback on our work. Your constructive feedback has been valuable in improving our work, and we sincerely appreciate your commitment.

We have incorporated revisions (where needed) as blue text in the revised manuscript.  Please let us know if you have any further suggestions. We would be happy to continue the discussion to address any remaining concerns.

---

### Comment · Action_Editor_16ki · 2025-06-27

I am now starting the decision process. I would be curious to hear what the authors changed more broadly in response to my preliminary meta-review. @Authors: If you have time, you would greatly help with summarizing the changes, as well as to make it more transparent post-review.  Thank you!

---

> ### Author Response · Authors · 2025-06-30
> **Summary of changes**
>
> Dear Action Editor:
>
> Absolutely! We appreciate your time and the constructive feedback from the reviewers to improve our work. Below, we provide a summary of the changes based on the meta-review. All the changes are highlighted in blue in the revised manuscript submitted as part of our rebuttal.
>
> **Taxonomy**
>
> _1) Terminology (long vs short-term)_: We further clarified the distinction between short-term and long-term fairness by explicitly describing long-term fairness as a "spectrum" in the Introduction =. Specifically, we explain that it ranges from _single-step analyses_ or _feedback loops_ to _multi-step_, _sequential evaluations_ over different time horizons. This explicitly emphasizes that long-term fairness is not a binary concept but a continuum, and we updated the text (Page 1) accordingly to reflect this.
>
> _2) Concepts_:
>
> - In response to the request for clearer conceptual distinctions, we revised the draft to clarify the **definitions**, **scope**, & **modeling assumptions** underlying performative prediction, delayed impact, and dynamic modeling (see Sections 2.2, 6.1, & 6.2).
>
>    -  We clarified that **performative prediction** refers to a **model-induced distribution shift** framework where predictions **causally** influence the environment. We also highlight that it is typically **stateless** (the environment’s response at each step depends only on the current prediction, without explicit dependence on past states), in contrast to dynamic modeling, and note its growing relevance in fairness and strategic behavior research. (Section 6.1)
>    -  Delayed impact is distinguished by its focus on **gradual** (rather than immediately) effects over time **without** requiring **ongoing model-environment feedback**. It assumes simpler, one-step feedback mechanisms, making it more analytically tractable than performative prediction or dynamic modeling (Section 6.2).
>    -  We emphasize that **dynamic modeling** represents the temporal evolution of a system, where the current state depends on **both** past states and actions, as well as feedback from the environment or population. It captures both immediate and delayed effects, rich feedback loops, and complex **ongoing** interactions. This is the primary setting that the literature has focused on. (Sections 2.2, 4)
>
> _3) Representative works:_ Table 1 summarizes representative works mapped to our taxonomy (mitigation, notions). However, we highlight that papers in the literature often do not propose mitigation strategies for all fairness notions; in fact, some notions are proposed solely for characterizing longer-term unfairness without addressing mitigation.
>
> **Coverage**
>
> - _Strategic behavior/incentives:_ We highlight how they remain relatively underexplored and are not yet major themes in the long-term fairness literature. We have revised Section 2.2 (Other frameworks) to highlight their relevance and encourage further exploration from multiple angles (Notions, mitigations), including future work in Sections 3.5, 4.3
>
> - _Effort-Fairness:_ We clarified that although effort-based fairness definitions resemble static fairness in form, their motivation and interpretation are inherently dynamic and long-term since they explicitly acknowledge that individuals can adjust their qualifications or features over multiple decision cycles (hence accumulate). This perspective frames fairness as an ongoing process, capturing disparities that persist or compound across repeated interactions with the system. We revise Section 3.1 to reflect this/
>
> - _Combinations of taxonomy:_ We recognize the possible confusion highlighted by the reviewer. We add a brief clarification in Section 2.1 to address this.
>
> **Critical Analysis**
>
> - _Real-world challenges and Normative assumptions:_ In response to the reviewer’s concern about practical feasibility, we clarified that while causal fairness methods face real-world challenges, developments in causal discovery provide "identification" guarantees under mild assumptions, supporting their applicability. We also addressed the normative assumption that certain decisions are inherently “positive” by adding a brief note in Section 7, highlighting that such implications should be explicitly considered in long-term fairness discussions, while maintaining fidelity to how these assumptions are treated in the original literature.
>
> - _Theoretical Results:_ We avoid adding theoretical results since it might risk making the survey a bit overly complex. We hope our survey is a foundational resource for those new to the field, which guided our design
>
> _A general note on Critical Analysis_: Since the meta-review was not visible earlier, we focused on addressing specific reviewer suggestions (e.g., causal challenges). We are happy to incorporate a broader critical analysis section and address the additional meta-review comments in the manuscript (including typos and suggested research agendas).
>
> We welcome any further suggestions you may have!

---

### Decision · Action_Editor_16ki · 2025-06-30

**Recommendation:** Accept with minor revision

**Additional Comments:**

Overall, some (minor) key issues remain that can be fixed a revision.

First, I noted specific suggestions regarding typesetting conventions in my meta-review below (e.g., consistent use of \citet vs. \citep) but which were not fully implemented throughout the manuscript. These inconsistencies should be carefully addressed in the final version.

Second, while the revised manuscript shows clear improvements, the paper still could do an even stronger job in a critical analysis. This was mentioned by both the reviewers and myself, but overall I see little to no edits in Section 7. As such, I encourage the authors to expand and deepen this critical assessment (e.g., via the current section or a separate one named "Critical reflection" or "Practical considerations" -- I leave that choice to the authors). I strongly believe that the value of a survey not only stems from summarizing the literature, but, more importantly, from *how* it maps the literature and how the mapping is used to identify research priorities, etc. Also, I strongly encourage the authors to to think about a meaningful table or so to map the critical analysis (e.g., a table nicely navigating research gaps along different dimensions or domains).

**Audience:**

Yes

**Audience Explanation:**

This paper addresses an important and timely topic: the evolving landscape of **long-term** fairness in machine learning. It offers a broad survey of key literature, introduces a multi-dimensional taxonomy (notions, modeling, mitigation), and outlines future directions. The authors have made substantial efforts to respond to prior reviewer comments, and the revised manuscript reflects improvements in conceptual clarity, especially around the distinctions between performative prediction, delayed impact, and dynamic modeling (and what "long-term" is more broadly). Overall, I think that the paper makes a descent effort to map the emerging literature in this field and will help scholars summarize themes and identify research gaps.

**Claims And Evidence:**

Yes

**Claims Explanation:**

The arguments are clear and accurate as found by the reviewers. Overall, while the revised manuscript shows meaningful progress and covers a wide breadth of literature, there is still some room to improve the depth of critical engagement, because of which I recommend a minor revision. I thank the authors for their hard work so far.

As a more personal thought: I find the tables summarizing the literature are helpful and appreciated. They aid in navigating a complex field. Nevertheless, that said, I had originally hoped to see some conceptual visualizations of how different notions, models, and outcomes interrelate. Some other overview papers manage to do that -- I recognize that producing such a figure might be non-trivial in the current setting due to the complexity of notions etc. and therefore do not request it at this stage — but, for completeness, I wanted note that a well-executed visual could elevate the manuscript's utility for readers.